# LEARN GOAL-CONDITIONED POLICY WITH INTRINSIC MOTIVATION FOR DEEP REINFORCEMENT LEARNING

## ABSTRACT

It is of significance for an agent to learn a widely applicable and general-purpose policy that can achieve diverse goals including images and text descriptions. Considering such perceptually-specific goals, the frontier of deep reinforcement learning research is to learn a goal-conditioned policy without hand-crafted rewards. To learn this kind of policy, recent works usually take as the reward the non-parametric distance to a given goal in an explicit embedding space. From a different viewpoint, we propose a novel unsupervised learning approach named goal-conditioned policy with intrinsic motivation (GPIM), which jointly learns both an abstract-level policy and a goal-conditioned policy. The abstract-level policy is conditioned on a latent variable to optimize a discriminator and discovers diverse states that are further rendered into perceptually-specific goals for the goal-conditioned policy. The learned discriminator serves as an intrinsic reward function for the goal-conditioned policy to imitate the trajectory induced by the abstract-level policy. Experiments on various robotic tasks demonstrate the effectiveness and efficiency of our proposed GPIM method which substantially outperforms prior techniques.

## 1 INTRODUCTION

Reinforcement learning (RL) makes it possible to drive agents to achieve sophisticated goals in complex and uncertain environments, from computer games (Badia et al., 2020; Berner et al., 2019) to real robot control (Lee et al., 2018; Lowrey et al., 2018; Vecerik et al., 2019; Popov et al., 2017), which usually involves learning a specific policy for individual task relying on task-specific reward. However, autonomous agents are expected to exist persistently in the world and have the ability to solve diverse tasks. To achieve this, one needs to *build a universal reward function* and *design a mechanism to automatically generate diverse goals* for training. Raw sensory inputs such as images have been considered as common goals for agents to practice on and achieve (Watter et al., 2015; Florensa et al., 2019; Nair et al., 2018; 2019), which further exacerbates the challenge for designing autonomous RL agents that can deal with such perceptually-specific inputs.

Previous works make full use of a goal-achievement reward function as available prior knowledge (Pong et al., 2018), such as Euclidean distance. Unfortunately, this kind of measurement in original space is not very effective for visual tasks since the distance between images does not correspond to meaningful distance between states (Zhang et al., 2018). Further, the measure function is applied in the embedding space, where the representations of raw sensory inputs are learned by means of using a latent variable model like VAE (Higgins et al., 2017b; Nair et al., 2018) or using the contrastive loss (Sermanet et al., 2018; Warde-Farley et al., 2018). We argue that these approaches taking prior non-parametric reward function in original or embedding space as above may limit the repertoires of behaviors and impose manual engineering burdens (Pong et al., 2019).

In the absence of any prior knowledge about the measure function, standard unsupervised RL methods learn *a latent-conditioned policy* through the lens of empowerment Salge et al. (2014); Eysenbach et al. (2018); Sharma et al. (2019) or the self-consistent trajectory autoencoder (Co-Reyes et al., 2018; Hausman et al., 2018). However, the learned policy is conditioned on the latent variables rather than perceptually-specific goals. Applying these procedures to goal-reaching tasks, similar to parameter initialization or the hierarchical RL, needs an external reward function for the new tasks; otherwise the learned latent-conditioned policy cannot be applied directly to user-specified goals.

Different from previous works, a novel unsupervised RL scheme is proposed in this paper to learn goal-conditioned policy by jointly learning an extra abstract-level policy conditioned on latent variables. The abstract-level policy is trained to generate *diverse abstract skills* while the goal-conditioned policy is trained to efficiently achieve perceptually-specific goals that are rendered from the states induced by the corresponding abstract skills. Specifically, we optimize a discriminator in an unsupervised manner for the purpose of reliable exploration (Salge et al., 2014) to provide the intrinsic reward for the abstract-level policy. Then the learned discriminator serves as an intrinsic reward function for the goal-conditioned policy to imitate the trajectory induced by the abstract-level policy. In essence, *the abstract-level policy can reproducibly influence the environment, and the goal-conditioned policy perceptibly imitates these influences.* To improve the generalization ability of goal-conditioned policy in dealing with perceptually-specific inputs, a latent variable model is further considered in the goal-conditioned policy to disentangle goals into latent generative factors.

The main contribution of our work is an unsupervised RL method that can learn perceptually-specific goal-conditioned policy without the prior reward function for autonomous agents. We propose a novel training procedure for this model, which provides an universal and effective reward function for various perceptual goals, e.g. images and text descriptions. Furthermore, we introduce a latent variable model for learning the representations of high-dimensional goals, and demonstrate the potential of our model to generalize behaviors across new tasks. Extensive experiments and detailed analysis demonstrate the effectiveness and efficiency of our proposed method.

## 2 PRELIMINARIES

**RL:** An agent interacts with an environment and selects actions in RL so as to maximize the expected amount of reward received in the long run (Sutton & Barto, 2018), which can be modeled as a Markov decision process (MDP) (Puterman, 2014). An MDP is defined as a tuple $\mathcal{M} = (S, A, p, R, \gamma)$, where $S$ and $A$ are state and action spaces, $p(\cdot|s, a)$ gives the next-state distribution upon taking action $a$ in state $s$, $R(s, a, s')$ is a random variable representing the reward received at transition $s \xrightarrow{a} s'$, and $\gamma \in [0, 1)$ is a discount factor.

**Intrinsic Motivation:** RL with intrinsic motivation obtains the intrinsic reward by maximizing the mutual information between latent variables $\omega$ and agent's behaviors $\tau$: $\mathcal{I}(\omega; \tau)$, where the specific manifestation of $\tau$ can be an entire trajectory (Achiam et al., 2018), an individual state (Eysenbach et al., 2018) or a final state (Gregor et al., 2016); and the specific implementation includes *reverse* and *forward* forms (Campos et al., 2020). Please refer to Aubret et al. (2019) for more details.

**Disentanglement:** Given an observation $x$ with a dimension of $N$, VAE (Kingma & Welling, 2013; Higgins et al., 2017a) is a latent model that pairs a top-down encoder $q(z|x)$ with bottom-up decoder network $p(x|z)$ by introducing a latent factor $z$, where $dim(z) < N$. To encourage the inferred latent factor $z$ to capture the generative factor of $x$ in a disentangled manner, an isotropic unit Gaussian prior $p(z) \sim N(0; I)$ is commonly used to control the capacity of the information bottleneck (Burgess et al., 2018) by minimizing the KL divergence between $q(z|x)$ and $p(z)$.

## 3 THE METHOD

In this section, we firstly formalize the problem and introduce the framework. Secondly, we elaborate on the process of how to jointly learn the goal-conditioned policy and abstract-level policy. Thirdly, disentanglement is applied in our setting to improve the generalization ability.

### 3.1 OVERVIEW

Given perceptually-specific goal $g$, our objective is to learn a goal-conditioned policy $\pi_\theta(a|\tilde{s}, g)$ that inputs state $\tilde{s}$ and $g$ and outputs action $a$ as shown in Fig. 1. The abstract-level policy $\pi_\mu(a|s, \omega)$ takes as input state $s$ and a latent variable $\omega$ and outputs action $a$, where $\omega$ corresponds to diverse latent skills. The discriminator $q_\phi$ is firstly trained at the abstract-level for reliable exploration, then it provides the reward signal for the goal-conditioned policy to imitate the trajectory induced by the abstract-level policy. The abstract-level policy is able to generate diverse states $s$ that are further rendered as diverse perceptually-specific goal $g = Render(s)$. On this basis, $\pi_\theta(a|\tilde{s}, g)$

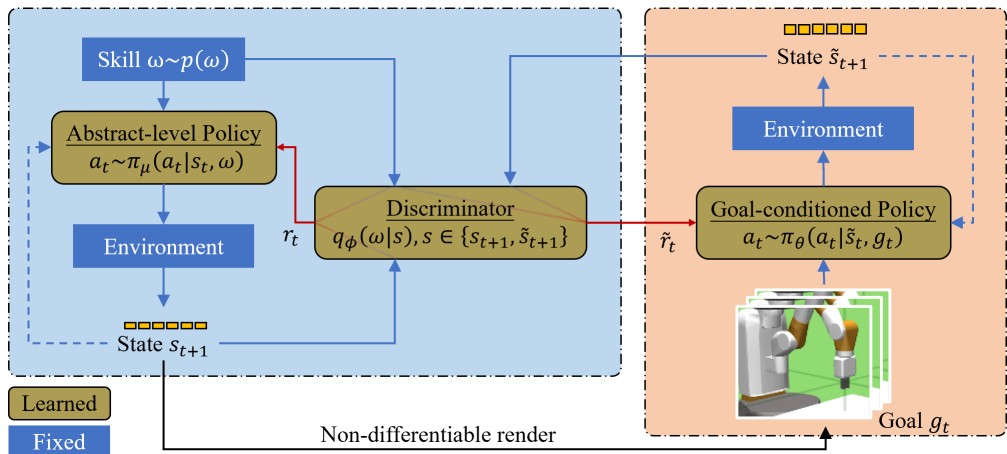

Figure 1: Framework of GPIM. We jointly train the abstract-level policy $\pi_\mu$ and the discriminator $q_\phi$ to understand skills which specify task objectives (e.g., trajectories, the final goal state), and use such understanding to reward the goal-conditioned policy for completing such (rendered) tasks. State $s_{t+1}$ (e.g., position of an agent) induced by $\pi_\mu$ is converted into a perceptually-specific goal $g_t$ (e.g., image showing the position of the agent) for $\pi_\theta$. Note that the two environments in the figure are identical, and the initial states $s_0$ of $\pi_\mu$ and $\tilde{s}_0$ of $\pi_\theta$ are sampled from the same distribution.

conditioned on the rendered goal $g$ interacts with the reset environment under the instruction of the reward function $q_\phi$. We use the non-tilde $s$ and the tilde $\tilde{s}$ to distinguish between the states of two policies respectively. Actually, $\tilde{s}$ and $s$ come from the same distribution.

## 3.2 PROPOSED GPIM METHOD

In order to jointly learn the abstract-level policy $\pi_\mu(a|s,\omega)$ and goal-conditioned policy $\pi_\theta(a|\tilde{s},g)$, we maximize the mutual information between the state $s$ and latent variable $w$ for $\pi_\mu$, and simultaneously maximize the mutual information between the state $\tilde{s}$ and goal $g$ for $\pi_\theta$. Consequently, the overall objective function to be maximized can be expressed as follows [1]

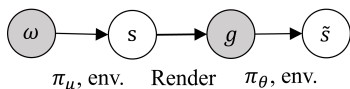

$$\pi_\mu, \text{env.} \quad \text{Render} \quad \pi_\theta, \text{env.}$$

Figure 2: Graphical model.

$$\mathcal{F}(\mu, \theta) = \mathcal{I}(s;\omega) + \mathcal{I}(\tilde{s};g). \tag{1}$$

For clarification, Fig. 2 depicts the graphical model for the latent variable $\omega$, state $s$ induced by $\pi_\mu$, goal $g$ rendered from $s$, and state $\tilde{s}$ induced by $\pi_\theta$. As seen, the latent variable $\omega \sim p(\omega)$[2] is firstly used to generate state $s$ via the policy $\pi_\mu$ interacting with dynamic environment. Then, we use the state $s$ to generate perceptually-specific goal $g$. After that, the goal-conditioned policy $\pi_\theta$ outputs action $a$ to interact with the environment to obtain the state $\tilde{s}$ at the next time step. In particular, $\omega$ is expected to generate diverse behavior modes through $\pi_\mu$ while $\pi_\theta$ behaving like $\tilde{s}$ is to imitate this behavior by taking as input the rendered goal $g$. Based on the context, the correlation between $\tilde{s}$ and $g$ is no less than that between $\tilde{s}$ and $w$, so that $\mathcal{I}(\tilde{s};g) \geq \mathcal{I}(\tilde{s};\omega)$ (Beaudry & Renner, 2011). Thus, we can obtain the lower bound of $\mathcal{F}(\mu, \theta)$:

$$\begin{aligned} \mathcal{F}(\mu, \theta) &\geq \mathcal{I}(s;\omega) + \mathcal{I}(\tilde{s};\omega) \\ &= 2\mathcal{H}(\omega) - \mathcal{H}(\omega|s) - \mathcal{H}(\omega|\tilde{s}) \\ &= \mathbb{E}_{\omega,s,\tilde{s}}\left[\log p(\omega|s) + \log p(\omega|\tilde{s}) - 2\log p(\omega)\right]. \end{aligned} \tag{2}$$

Since it is difficult to exactly compute the posterior distributions $p(\omega|s)$ and $p(\omega|\tilde{s})$, Jensen's Inequality (Barber & Agakov, 2003) is further applied for approximation by using a learned discrimi-

---

[1]To further clarify the motivation, we conduct the ablation study to compare our method with that just maximizing $\mathcal{I}(s;\omega)$ and that just maximizing $\mathcal{I}(s;g)$ in Appendix A.1.

[2]$p(\omega)$ denotes the prior distribution. Fixing $p(\omega)$ instead of learning it is to prevent $\pi_\mu$ from collapsing to sampling only a handful of skills. In experiment, we use a categorical distribution for $p(\omega)$ following DIAYN.

---

**Algorithm 1** Learning process of our proposed GPIM

1: **while** not converged **do**
2:     Sample skill $\omega \sim p(\omega)$.
3:     *# Step I*
4:     Sample initial state $s_0 \sim p_0(s)$.
5:     **for** $t = 0, 1, ..., T$ steps **do**
6:         Sample action $a_t \sim \pi_\mu(a_t|s_t, \omega)$.
7:         Step environment: $s_{t+1} \sim p(s_{t+1}|s_t, a_t)$.
8:         Render goal: $g_t = Render(s_{t+1})$.
9:         Compute reward $r_t$ for policy $\pi_\mu$ using (5).
10:        Update policy $\pi_\mu$ to maximize $r_t$ with SAC.

11:     Update discriminator ($q_\phi$) to maximize $\log q_\phi(\omega|s_{t+1})$ with SGD.
12:     **end for**

13:     *# Step II*
14:     Sample initial state $\tilde{s}_0 \sim p_0(\tilde{s})$.
15:     **for** $t = 0, 1, ..., T - 1$ steps **do**
16:         Sample action $a_t \sim \pi_\theta(a_t|\tilde{s}_t, g_t)$.
17:         Step environment: $\tilde{s}_{t+1} \sim p(\tilde{s}_{t+1}|\tilde{s}_t, a_t)$.
18:         Compute reward $\tilde{r}_t$ for policy $\pi_\theta$ using (7).
19:         Update policy $\pi_\theta$ to maximize $\tilde{r}_t$ with SAC ($\theta = \{\vartheta_E, \vartheta_G\}$ when considering the disentanglement).
20:        Update $\vartheta_E$ and $\vartheta_D$ by maximizing *Dis_loss* in (8) using SGD.
21:     **end for**
22: **end while**

---

nator network $q_\phi(\omega|\cdot)$ (see Appendix B.1 for the derivation). Thus,

$$\mathcal{F}(\mu, \theta) \geq \mathbb{E}_{\omega, s, \tilde{s}} \left[ \log q_\phi(\omega|s) + \log q_\phi(\omega|\tilde{s}) - 2 \log p(\omega) \right] \triangleq \mathcal{J}(\mu, \phi, \theta), \tag{3}$$

where it is worth noting that the identical discriminator $q_\phi$ with the parameter $\phi$ is used for the variational approximation of $p(\omega|s)$ and $p(\omega|\tilde{s})$. For the state $s$ induced by skill $w$ and $\tilde{s}$ originating from $g$, the shared discriminator $q_\phi$ assigns a similarly high probability on $w$ for both states $s$ and $\tilde{s}$ associated with the same $\omega$. Intuitively, we factorize the abstract-level policy and learn it purely in the space of the agent's embodiment (i.e., the latent variable $\omega$) — separate from the perceptually-specified goals (e.g., images and text descriptions). The space of goals in these two spaces has different characteristics due to the underlying manifold spaces. Therefore, $q_\phi$ can be regarded as a reward network shared by the abstract-level policy $\pi_\mu(a|s, \omega)$ and goal-conditioned policy $\pi_\theta(a|\tilde{s}, g)$.

According to the objective function $\mathcal{J}(\mu, \phi, \theta)$ in (3), we propose an alternating optimization between $\pi_\mu$ and $\pi_\theta$ as follows:

**Step I:** Fix $\pi_\theta$ and update $\pi_\mu$ and $q_\phi$. In this case, $\theta$ is not a variable to update and $\mathcal{J}(\mu, \phi, \theta)$ becomes

$$\mathcal{J}(\mu, \phi) = \mathbb{E}_{\omega, s} \left[ \log q_\phi(\omega|s) \right] + \underbrace{\mathbb{E}_{\omega, \tilde{s}} \left[ \log q_\phi(\omega|\tilde{s}) - 2 \log p(\omega) \right]}_{\text{Variable independent term}}. \tag{4}$$

According to (4), $\mathcal{J}(\mu, \phi)$ can be thus optimized by setting the reward at time step $t$ for $\pi_\mu$ as

$$r_t = \log q_\phi(\omega|s_{t+1}) - \log p(\omega), \tag{5}$$

where the term $-\log p(\omega)$ is added for agents to avoid artificial termination and reward-hacking issues (Amodei et al., 2016). We implement this optimization with soft actor-critic (SAC). On the other hand, the reward network $q_\phi$ can be updated with SGD by maximizing $\mathbb{E}_{\omega, s} [\log q_\phi(\omega|s)]$.

**Step II:** Fix $\pi_\mu$ and $q_\phi$ to update $\pi_\theta$. In this case, $\mu$ and $\phi$ are not variables to update any more and $\mathcal{J}(\mu, \phi, \theta)$ can be simplified as

$$\mathcal{J}(\theta) = \mathbb{E}_{\omega, \tilde{s}} \left[ \log q_\phi(\omega|\tilde{s}) \right] + \underbrace{\mathbb{E}_{\omega, s} \left[ \log q_\phi(\omega|s) - 2 \log p(\omega) \right]}_{\text{Variable independent term}}. \tag{6}$$

According to (6), $\mathcal{J}(\theta)$ can thus be optimized by setting the reward at time step $t$ for $\pi_\theta$ as

$$\tilde{r}_t = \log q_\phi(\omega|\tilde{s}_{t+1}) - \log p(\omega), \tag{7}$$

where the term $-\log p(\omega)$ is added for the same reason as above and we also implement this optimization with SAC. These two steps are performed alternately until convergence. The overall GPIM is summarized in Algorithm 1, which includes the disentanglement (*red*) in the next subsection.

## 3.3 IMPROVE GENERALIZATION VIA DISENTANGLEMENT

The goal $g_t$ can be further disentangled (Higgins et al., 2017a) so as to improve the generalization ability of the goal-conditioned policy $\pi_\theta$.

As shown in Fig. 3, we decompose $\pi_\theta$ into two components: the encoder network E parameterized by $\vartheta_E$ and the generative network G parameterized by $\vartheta_G$. So we have $\pi_\theta(a_t|\tilde{s}_t, g_t) = p_{\vartheta_G}(a_t|\tilde{s}_t, q_{\vartheta_E}(z|g_t))$.

To encourage this disentangling property in the inferred $q_{\vartheta_E}(z|g_t)$, we adopt an prior $p(z) \sim N(0; I)$ to control the capacity of the information bottleneck by minimizing the KL divergence between $q_{\vartheta_E}(z|g_t)$ and $p(z)$. Hence, the overall objective function of goal-conditioned policy $\pi_\theta$ becomes

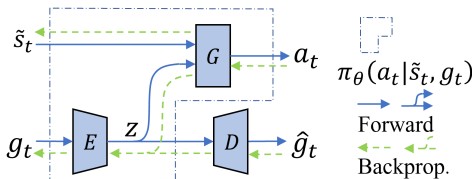

Figure 3: Disentanglement of policy $\pi_\theta$.

$$\max_{\vartheta_E,\vartheta_G,\vartheta_D} \mathbb{E}_{\omega,\tilde{s}_t}\left[\tilde{r}_t\right] + \underbrace{\alpha \cdot \mathbb{E}_{g_t,z}\left[p_{\vartheta_D}(g_t|z)\right] - \beta \cdot \mathbb{E}_{g_t}\left[KL(q_{\vartheta_E}(z|g_t)||p(z))\right]}_{\triangleq Dis\_loss}, \qquad (8)$$

where $\alpha$ and $\beta$ are two hyperparameters. It is worth noting that the update of the encoder E is based on gradients from both $\pi_\theta$ and *Dis_loss*. With this disentanglement, the overall GPIM method is given in Algorithm 1 as mentioned above.

## 4 RELATED WORK

*Investigating goal distribution:* Many prior methods (Schaul et al., 2015; Andrychowicz et al., 2017; Pong et al., 2018; Hartikainen et al., 2019) assume an available distribution of goals during exploration for sparse-reward problems, where the resampled goals are restricted to states encountered along the trajectory (Andrychowicz et al., 2017; Levy et al., 2017) or in an embedding space (Nair et al., 2018). The setting of goals for learning policy has been discussed in Baranes & Oudeyer (2013). Recently, several works (Colas et al., 2018; Warde-Farley et al., 2018; Florensa et al., 2019; Nair et al., 2018; Péré et al., 2018) adopt heuristics to design a goal distribution based on previously visited states. However, many of these (Nair et al., 2018; Pong et al., 2019; Warde-Farley et al., 2018) assume an ideal matching of the goal space with the state space.

*Learning goal-conditioned policy:* Several recent works propose various methods for agents to automatically learn policies by maximizing the mutual information between behaviors (e.g., latent variables) and tasks (e.g., goals). DISCERN (Warde-Farley et al., 2018) simultaneously learns a goal-conditioned policy and a goal achievement reward function that measures how similar a state is to the goal state. Pong et al. (2019) concurrently trains a goal-reaching policy and maximizes the entropy of the generated goal distribution. Given the difficulty of vision-based RL, Pong et al. (2019) resorts to RIG (Nair et al., 2018), which obtains reward using prior non-parametric measure function in embedding spaces. Other unsupervised methods (Sharma et al., 2019; Campos et al., 2020), capable of learning diverse skills, are difficult to apply directly to user-specified tasks.

*Learning disentanglement representation:* Many prior works utilize unsupervised learning in RL to acquire disentanglement representation (Bengio et al., 2013; Higgins et al., 2017a; Chen et al., 2016; Burgess et al., 2018). Some works take the disentanglement representation as a substitute for states (Higgins et al., 2017b; Ha & Schmidhuber, 2018; Jonschkowski et al., 2017; Finn et al., 2016; Lange et al., 2012; Watter et al., 2015; Srinivas et al., 2018; Nair et al., 2019; Srinivas et al., 2018) while others consider a non-parametric metric in a latent space to acquire a reward function (Nair et al., 2018; Sermanet et al., 2018). Besides, a few works directly embed manipulation tasks to generalize to long-horizon tasks (Goyal et al., 2019; Hausman et al., 2018; Mandlekar et al., 2020).

*Hindsight, self-play and knowledge distillation:* Our method is similar in spirit to goal relabeling methods like hindsight experience replay (HER) (Andrychowicz et al., 2017) which replays each episode with a different goal in addition to the one the agent was trying to achieve. However, HER requires a prior reward function and a hand-crafted goal space. By contrast, GPIM is unsupervised and able to find its own goal space. The self-play (Sukhbaatar et al., 2017; 2018) is closely related to our scheme, leading to emergent autocurricula by pitting two versions of the same agent against one another. However, these approaches generally require prior reward functions. As for knowledge distillation (Xu et al., 2020), we aim at extracting the relationship between two different tasks.

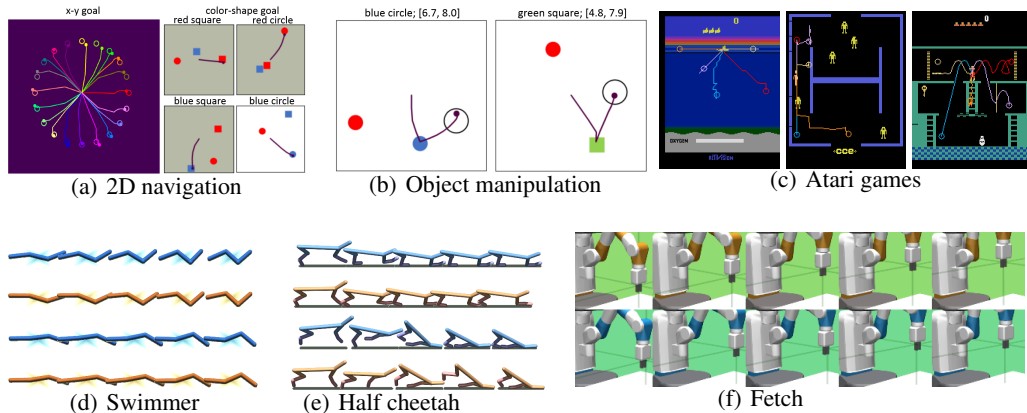

(a) 2D navigation     (b) Object manipulation     (c) Atari games

(d) Swimmer     (e) Half cheetah     (f) Fetch

Figure 5: Goals and learned behaviors by GPIM: Dots in 2D navigation (left subfigure, x-y goal) and atari games denote different final goal states, and curves with same color represent corresponding trajectories; Goals in 2D navigation (right subfigure, color-shape goal) and object manipulation are described using the text at the top of the diagram, where the purple lines imply the behaviors; In mujoco tasks, the first (swimmer, half cheetah and fetch) and third (swimmer and half cheetah) rows represent the expert trajectories, and each row below represents the corresponding behavior.

## 5 EXPERIMENTS

Extensive experiments are conducted to evaluate our proposed GPIM method, where the following four questions[3] will be considered in the main paper: 1) By using the "archery" task, we clarify whether $q_\phi$ can provide an effective reward function on learning the goal-conditioned policy $\pi_\theta$. Furthermore, more complex tasks including navigation, object manipulation, atari games, and mujoco tasks are introduced to answer: 2) Does our model learn effective behaviors conditioned on a variety of goals, including high-dimensional images and text descriptions that are heterogeneous to states? 3) Does the proposed GPIM on learning a goal-conditioned policy outperform baselines? 4) Does the learned reward function produce better expressiveness of tasks, compared with the prior non-parametric function in the embedding space? Video is available under https://sites.google.com/view/gpim.

**Visualizing the learned reward function.** We start with simple "archery" task to visualize how the learned reward function (discriminator $q_\phi$) accounts for goal-conditioned behaviors in dynamics. The task shown in Fig. 4 requires choosing an angle at which we shoot an arrow to the target. The left upper subfigure shows that in a deterministic environment, given three different but fixed targets (with different colors), the arrow reaches the corresponding target successfully under the learned reward function

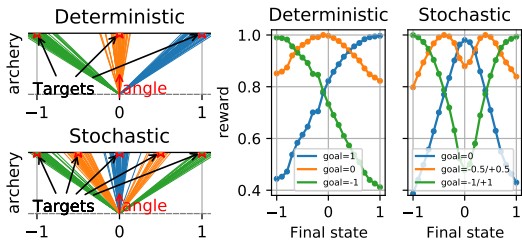

Figure 4: "Archery" tasks and learned rewards.

$q_\phi$. The reward as a function of the final location of arrows in three tasks is shown on the right. We can find that the learned reward functions resemble convexes in terms of the distance between final states to targets, where vertexes appear in the positions of the corresponding targets. Specifically, the maximum value of the learned reward function is achieved when the final state is close to the given target. The farther away the agent's final state is from the target, the smaller this reward value is. Similarly, the same conclusion can be drawn from the stochastic environment in the left lower subfigure, where the angle of the arrow has a 50% chance to become a mirror-symmetric angle. We see that the learned reward function substantially describes the dynamics and the corresponding tasks, both in deterministic and stochastic environments. This answers our *first* question.

---

[3]Due to the page limitation, more experimental analysis and results are included in the Appendix.

**Scaling to more complex tasks.** To answer our *second* question, we now consider more complex tasks as shown in Fig. 5. 1) In *2D navigation tasks*, an agent can move in each of the four cardinal directions. We consider the following two tasks: moving the agent to a specific coordinate named *x-y goal*[4] and moving the agent to a specific object with certain color and shape named *color-shape goal*. 2) *Object manipulation* considers a moving agent in 2D environment with one block for manipulation, and the other block as a distractor. The agent first needs to reach the block and then move the block to the given location, where the block is described using color and shape. In other words, the description of the goal contains the *color-shape goal* of the true target block and the *x-y goal* of the target coordinate. 3) Three *atari games* including seaquest, berzerk and montezuma revenge require an agent to reach the given final states. 4) We use three *mujoco tasks* (swimmer, half cheetah, and fetch) taken from OpenAI GYM (Brockman et al., 2016) to fast imitate given expert trajectories. Specifically, the goals for $\pi_\theta$ in 2D navigation, object manipulation and atari games are the rendered final state $s_T$ induced by abstract-level policy $\pi_\mu$: $g_t = Render(s_T)$, and the goals for $\pi_\theta$ in mujoco tasks are the rendered trajectories induced by $\pi_\mu$: $\{g_0, g_1, ...\} = Render(\{s_1, s_2, ...\})$.

The left subfigure of Fig. 5(a) shows the learned behaviors of navigation in continuous action space given the x-y goal which is denoted as the small circle, and the right subfigure shows the trajectory of behavior with the given color-shape goal. More results on discrete navigation can be found in Appendix D.2. As observed, the agent manages to learn navigation tasks by using GPIM. Further, 2D navigation with color-shape goal (Fig. 5(a) *right*) and object manipulation tasks (Fig. 5(b)) show the effectiveness of our model facing heterogeneous goals and states. Specifically, Fig. 5(b) shows the behaviors of the agent on object manipulation, where the agent is asked to first arrive at a block (i.e., blue circle and green square respectively) and then push it to the given location inside a dark circle (i.e., [6.7, 8.0] and [4.8, 7.9] respectively), where the red object exists as a distractor. Fig. 5(c) shows the behaviors of agents that reach the final states in a higher dimensional (action, state and goal) space on seaquest, berzerk and montezuma revenge respectively. Fig. 5(d-f) shows how the agent imitates expert trajectories of swimmer, half cheetah and fetch.

By learning to reach diverse goals generated by the abstract-level policy and then disentangling the goals, the agent learns the ability to infer new goals later encountered by the agent. For example, as in Fig. 5(a) (*right*), learning three behaviors with the goal of red-square, red-circle or blue-square in a gray background makes the agent accomplish the new goal of blue-circle in a white background. We further conduct the ablation study in Appendix A.3 to show how the disentanglement affects the learned behaviors, and use more experiments to show the generalization of the goal-conditioned policy to unseen goals in Appendix A.4.

Solving complex temporally-extended tasks is a long-standing RL problem (Jiang et al., 2019). For this purpose, we further make agents imitate the given composite behaviors (i.e., expert behaviors) and show their performance in Fig. 6. Specifically, we consider the fetch task as in Fig. 5(f), where the gripper of the robot arm is used to render a series of 3D coordinates as goals for imitation in training phase. During test, we employ two parameterized complex

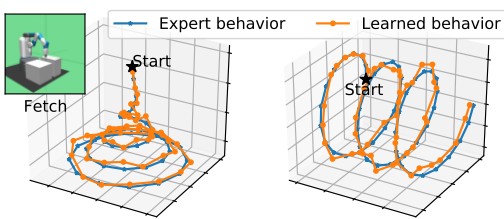

Figure 6: Expert behaviors and learned behaviors.

curves for the gripper to follow: $(x, y, z) = (t\sin(t)/50, t\cos(t)/50 - 1/50, -\log(t+1)/5)$ (left) and $(x, y, z) = (t/5, \cos(t)/5 - 1/5, \sin(t)/5)$ (right). It is worth noting that during training the agent is required to imitate a large number of simple behaviors and has never seen such complex goals before testing. It is observed from Fig. 6 that the imitation curves are almost overlapping with the given expert trajectories, indicating that the agent using GPIM has the potential to learn such compositional structure of goals during training and generalize to new composite goals during test.

**Comparison with baselines.** For the *third* question, we compare our method to three baselines: **RIG** (Nair et al., 2018), **DISCERN** (Warde-Farley et al., 2018), and **L2 Distance**. L2 Distance measures the distance between states and goals, where the $L2$ distance $-||s_t - s_g||^2/\sigma_{pixel}$ is considered with a hyperparameter $\sigma_{pixel}$. Note that 2D navigation with the color-shape goal and object manipulation using text description makes the dimensions of states and goals different, so L2 cannot be used in these two tasks. In RIG, we obtain rewards by using the distances in two embedding

---

[4]Definitions of *x-y goal*, *color-shape goal*, and environment details are given in Appendix B.2.

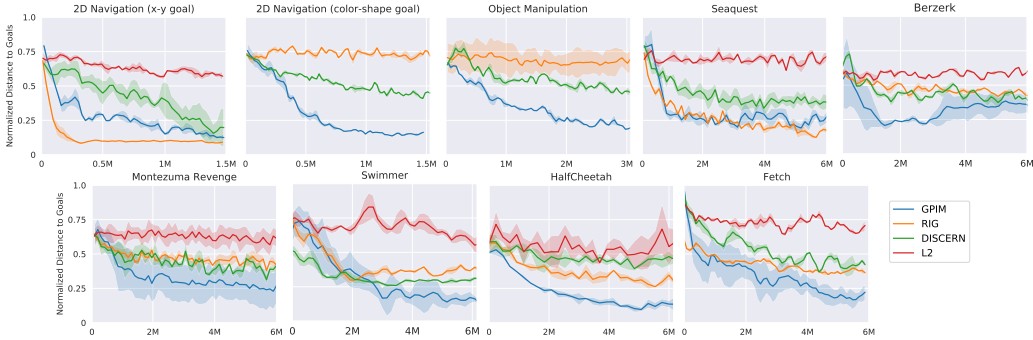

Figure 7: Performance (normalized distance to goals vs. training steps) of our GPIM and three baselines, where GPIM generally achieves smaller distances to goals in comparison to baselines.

spaces and learning two independent VAEs, where one VAE is to encode states and the other is to encode goals. We use the normalized distance to goals [5] as the evaluation metric, where we randomly sampled 50 samples (tasks) from the goal space.

We show the results in Fig. 7 by plotting the normalized distance to goals as a function of the number of actor's steps, where each curve considers 95% confidence interval in terms of the mean value across three seeds. As observed, our GPIM consistently outperforms baselines in almost all tasks except for the RIG in 2D navigation (x-y goal) due to the simplicity of this task. Particularly, as the task complexity increases from 2D navigation (x-y goal) to 2D navigation (color-shape goal) and eventually object manipulation (mixed x-y goal and color-shape goal), GPIM converges faster than baselines and the performance gap between our GPIM and baselines becomes larger. Moreover, although RIG learns fast on navigation with x-y goal, it fails to accomplish complex navigation with color-shape goal because the embedding distance between two independent VAEs has difficulty in capturing the correlation of heterogeneous states and goals. Especially in high-dimensional action space and on more exploratory tasks (atari and mujoco tasks), our method substantially outperforms the baselines.

To gain more intuition for our method, we record the distance ($\Delta r$) between the final state induced by $\pi_\theta$ and the goal rendered by $\pi_\mu$ throughout the training process of the 2D navigation (x-y goal). For this purpose, in this specific experiment, we update $\pi_\mu$ and $q_\phi$ but ignore the update of $\pi_\theta$ before $200\,\text{k}$ steps to show the exploration of $\pi_\mu$ at the abstract level. As shown in Fig. 8, $\Delta r$ steadily increases during the first $200\,\text{k}$ steps, indicating that the abstract-level policy $\pi_\mu$ explores the environment (i.e., goal space) to distinguish skills more easily,

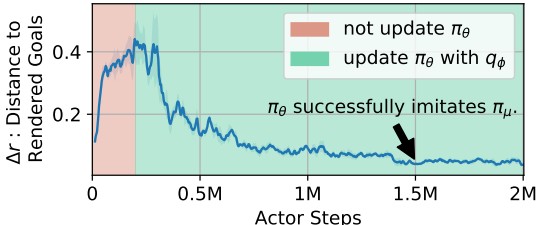

Figure 8: Abstract-level policy $\pi_\mu$ gradually explores the environment, generating more difficult goals. $q_\phi$ encourages $\pi_\theta$ to gradually mimic $\pi_\mu$.

and as a result, generates diverse goals for training $\pi_\theta$. After around $1.5\,\text{M}$ training steps, $\Delta r$ almost comes to 0, indicating that the goal-conditioned policy $\pi_\theta$ has learned a good strategy to reach the rendered goals. In Appendix A.2, we visually show the generated goals at the abstract level in more complex tasks, which shows that our straightforward framework can effectively explore the environment without additional sophisticated exploration strategies.

**Expressiveness of the reward function.** Particularly, the performance of unsupervised RL methods depends on the diversity of autonomously generated goals and the expressiveness of the learned reward function, which is conditioned on the generated goals. We have shown that our straightforward framework can effectively explore the environment, achieving competitive performance with baselines (see appendix A.2). The next question is that: with the same exploration capability to generate goals for training, does our model achieve competitive performance against the baselines? Said another way, will the obtained reward (over embedding space) of baselines taking the prior nonparametric function limit the repertoires of learning tasks in a specific environment? Our next exper-

---

[5]Definition and further descriptions can be found in Appendix B.3.

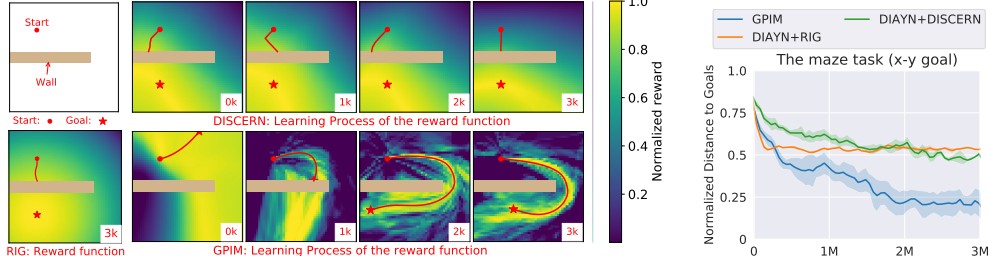

Figure 9: (Left) The maze environment and reward functions. The heatmaps depict (the learning process of) the reward function conditioned on the specific task reaching the left-bottom star (RIG and DISCERN) or "imitating" the trajectory induced by abstract-level policy (GPIM). Specifically, the learning process of DISCERN's reward function refers to the learning process of embedding state. Note that the reward functions of baselines are conditioned on the goals, while GPIM's reward function is conditioned on the skill $\omega$. So, the induced trajectory by GPIM conditioned on the same skill *refines* over training steps (at the abstract-level), as shown in the bottom. (Right) Learning curves for GPIM and the enhanced baselines (DIAYN+RIG and DIAYN+DISCERN). Compared with our model, baselines ignoring the dynamic of the maze environment exhibit poor performance.

iment studies the expressiveness of the learned reward function. For better graphical interpretation and comparison with baselines, we simplify the complex Atari games to a maze environment shown in Fig. 9, where the middle wall poses a bottleneck state. Campos et al. (2020) shows that the canonical information-theoretic skill discovery methods suffer from a poor coverage of the state space. Here, borrowing the idea from state marginal matching (Lee et al., 2019), we set the reward for the abstract-level policy as (Jabri et al., 2019) $r'_t = \lambda \left[ \log q_\phi(\omega|s_{t+1}) - \log p(\omega) \right] + (\lambda - 1) \log q_\nu(s_{t+1})$, where $q_\nu$ is a density model, and $\lambda \in [0, 1]$ can be interpreted as trade off between discriminability of skills and task-specific exploration (here we set $\lambda = 0.5$). Note that we modify $r'_t$ for improving the exploration on generating goals, and we do not change the reward for training the goal-conditioned policy $\pi_\theta$. To guarantee generation of the same diverse goals for training goal-conditioned policies of baselines, we adopt DIAYN taking the modified reward $r'_t$ to generate goals for RIG and DISCERN, denoted as DIAYN+RIG and DIAYN+DISCERN respectively.

In Fig. 9, we show the visualized learned reward on a specific task reaching the left-bottom star, and the learning curves on the maze task, where the testing-goals are random sampled. We can see that the learned reward functions of RIG and DISCERN produce poor signal for the goal-conditioned policy, which makes learning vulnerable to local optima. Our method builds up the reward function after exploring the environment, the dynamic of which itself further shapes the reward function. In Fig. 9 (left), we can see that our model provides the reward function better expressiveness of the task by compensating for the dynamic. This produces that, even with the same exploration capability to generate diverse goals, our model sufficiently outperforms the baselines, as shown in Fig. 9 (right).

## 6 CONCLUSION

We propose a novel GPIM method to learn goal-conditioned policy in an unsupervised manner. Specifically, we optimize a discriminator in an unsupervised manner for the purpose of reliable exploration to provide the intrinsic reward for the abstract-level policy. The learned discriminator then serves as an intrinsic reward function for the goal-conditioned policy to imitate the trajectory induced by the abstract-level policy. Experiments on a variety of robotic tasks demonstrate the effectiveness and efficiency of our proposed method which substantially outperforms prior unsupervised techniques.

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

# A ADDITIONAL EXPERIMENTS

## A.1 COMPARISON WITH DIAYN AND ITS VARIANT

In this section, we expect to clarify the difference and connection with DIAYN (Eysenbach et al., 2018) experimentally, and indicate the limitations of maximizing $\mathcal{I}(s;\omega)$ and maximizing $\mathcal{I}(s;g)$ separately on learning goal-conditioned policy.

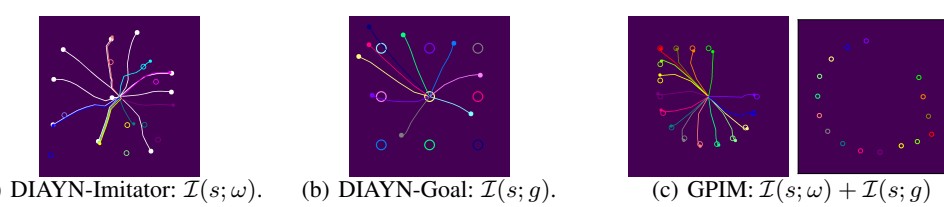

(a) DIAYN-Imitator: $\mathcal{I}(s;\omega)$.  (b) DIAYN-Goal: $\mathcal{I}(s;g)$.  (c) GPIM: $\mathcal{I}(s;\omega)+\mathcal{I}(s;g)$

Figure 10: 2D navigation with (a) DIAYN-Imitator, (b) DIAYN-Goal, and (c) our proposed GPIM. In (a), (b) and (c-left), the open circle denotes the user-specified goal, and the line with a solid circle at the end is the corresponding behavior. Specifically, the white lines in (a) are the learned skills in training, where we show 10 diverse skills. We can see that neither DIAYN-Imitator nor DIAYN-Goal approaches the corresponding goals, while our model succeeds in reaching the goal as shown in (c-left). In (c-right), we also show the generative factor $z$ by disentangling goals in (c-left), where the open circles with different colors in (c-right) correspond to the goals in (c-left).

In DIAYN, authors show the ability of the model to imitate an expert. Given the goal, DIAYN uses the learned discriminator to estimate which skill was most likely to have generated the goal $g$:

$$\hat{\omega} = \arg\max_{\omega} q_{\phi}(\omega|g). \tag{9}$$

Here we call this model *DIAYN-Imitator*. We also directly substitute the perceptually-specific goals for the latent variable in DIAYN's objective to learn a goal-conditioned policy. We call this model *DIAYN-Goal*:

$$\max \ \mathcal{I}(s;g), \tag{10}$$

where $g$ is sampled from the prior goal distribution $p(g)$. Please note that we do not adopt the prior non-parametric distance as in DISCERN (Warde-Farley et al., 2018) to calculate the reward. We obtain the reward as in normal DIAYN using the variational inference: $r_t = q_{\phi}(g_t|s_{t+1})$.

Fig. 10 shows the comparison of our GPIM with DIAYN variants including DIAYN-Imitator and DIAYN-Goal, where the 2D navigation task is considered. As observed, DIAYN-Imitator can reach seen goals but not unseen goals in Fig. 10(a), because it cannot effectively accomplish the interpolation between skills that are induced in training. And behaviors generated by DIAYN-Goal cannot guarantee consistency with the preset goals in Fig. 10(b). The main reason is that this objective only ensures that when $g$ (or $\omega$) is different, the states generated by $g$ (or $\omega$) are different. However, there is no guarantee that $g$ (or $\omega$) and the state generated by the current $g$ (or $\omega$) have semantically consistent behavior information. Our proposed GPIM method, capable of solving interpolation and consistency issues, exhibits the best performance in this 2D navigation task.

Moreover, when the user-specified goals are heterogeneous to the states, the learned discriminator $q_{\phi}$ in DIAYN is unable to estimate which skill is capable of inducing given goals. Specifically, when the goals are visual inputs, and the states in training is feature vectors (e.g., joint angles), the learned discriminator is unable to choose the skills due to a lack of models for converting high-dimensional figure into low-dimensional feature vectors. On the contrary, there are lots of off-the-shelf models to render low-dimensional feature vectors into perceptually-specific high-dimensional inputs (Tobin et al., 2017; Khirodkar et al., 2018).

## A.2 AUTOMATED GOAL-GENERATION FOR EXPLORATION

In general, for unsupervised RL, we would like to ask the agent to carry out autonomous "practice" during training phase, where we do not know which particular goals will be provided in test phase.

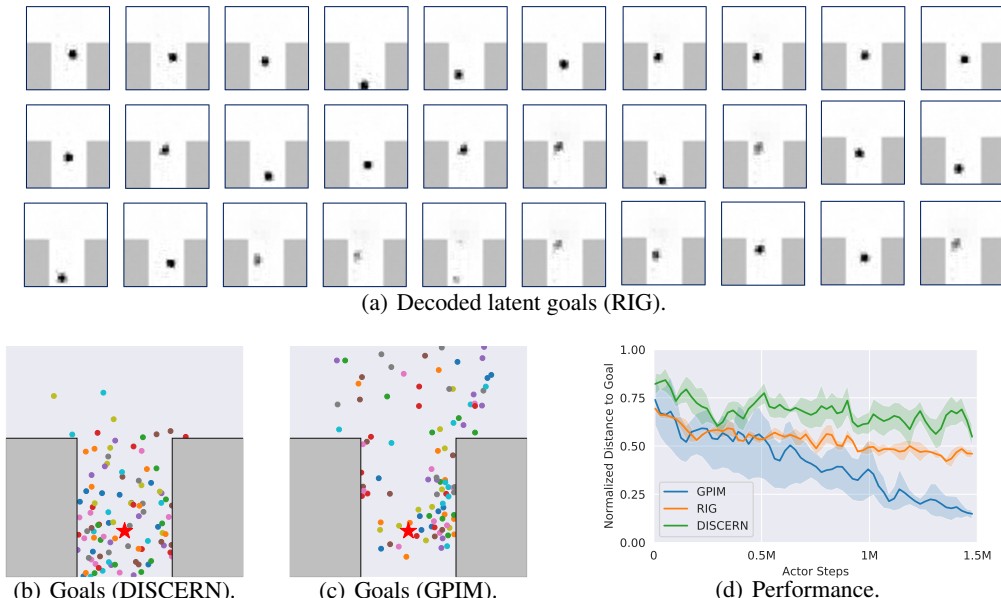

(a) Decoded latent goals (RIG).

(b) Goals (DISCERN).    (c) Goals (GPIM).    (d) Performance.

Figure 11: Distribution of sampled goals for the goal-conditioned policy, where the initial state is [5,2] (the red star) and the actor step is 20 for each rollout. (a) The goals (images) are obtained by decoding 30 sampled latent goals $z_g$ in VAE framework; (b) The goals (colored dots) are sampled from the agent's behaviors by random exploration; (c) The goals (colored dots) are rendered from the states induced by our abstract-level policy. (d): Evaluation on reaching user-specified goals, where GPIM significantly outperforms baselines.

In order to maximize the state coverage, the ability to automatically explore the environment and discover diverse goals is crucial. In this section, we will further analyze the goal distribution of three methods (RIG (Nair et al., 2018), DISCERN (Warde-Farley et al., 2018), GPIM) in a new 2D obstacle navigation task as shown in Fig. 11(a), Fig. 11(b), and Fig. 11(c). The size of the environment is $10 \times 10$, the initial state is set as $[5, 2]$, and there are two obstacles that prevent the agent from passing through each of which is $3 \times 6$ in size.

DISCERN samples goals during training by maintaining a fixed sized buffer $\mathcal{G}$ of past observations. We simply mimic the process of goal generation by taking random actions for 20 environment steps after initialization of the method. As in Fig. 11(b), we generate 100 goals with different colors. We can see that the majority of goals locate between the two obstacles, which limits the further exploration of the environment.

RIG samples a representation (latent goals $z_g$) from the learned VAE prior, which represents a distribution over latent goals and state observation. The policy network takes the representation as a substitute for the user-specified goal. For a clear visualization of the distribution of the sampled latent goals, we further feed the sampled latent goals into the decoder to obtain the real goals in the user-specified space. The decoded latent goals are shown in Fig. 11(a), where we sample 30 goals. It is shown that the majority of goals are also between the two obstacles because the goals for training the VAE prior come from the same distribution as in DISCERN.

Our method, GPIM, obtains goals from the behaviors induced by the abstract-level policy. Maximizing $\mathcal{I}(s; \omega)$ encourages different skills to induce different states that are further rendered to goals. This objective ensures that each skill individually is distinct and the skills collectively explore large parts of the state space (Eysenbach et al., 2018). As shown in Fig. 11(c), our method provides better coverage of the state space than DISCERN and RIG.

Fig. 11(d) shows the performance of the three methods, where we randomly sample goals from the whole state (or goal) space at test phase. We can see that our method significantly outperforms the baselines. The most common failure mode for prior methods is that the goal distribution col-

lapses (Pong et al., 2019), causing that the agent can reach only a fraction of the state space, as shown in Fig. 11(a) and 11(b).

Exploration is a well-studied problem in the field of RL, and there are many proven approaches with different benefits to improve the exploration (Colas et al., 2018; Campos et al., 2020). Note that these benefits are orthogonal to those provided by our straightforward GPIM, and these approaches could be combined with GPIM for even greater effect. We leave combing our method with sophisticated exploration strategies to future work.

### A.3   ABLATION STUDY

Here we conduct the ablation study on 2D navigation task to analyze how the disentanglement of goals in GPIM affects the learned behaviors in terms of the generalization to new tasks?. For convenience, we remove certain component from GPIM and define the new method as follows: *w/o Rec* - removing the reconstruction loss (i.e. $\alpha = 0$); *w/o KL* - removing the KL loss (i.e. $\beta = 0$); *w/o Pol* - removing the policy loss updating $\vartheta_E$ as shown in Fig. 3.

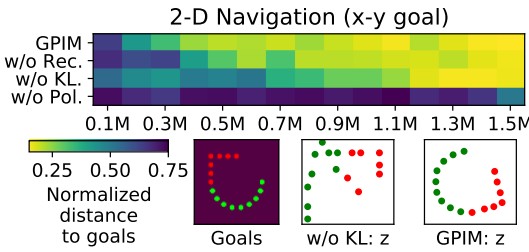

Figure 12: Ablation study and disentanglement.

The performance is reported in Fig. 12. It is observed that *w/o. Pol* performs worse than all other methods, which is consistent with the performance of RIG that trains VAE and policy separately. The main reason is that the disentanglement fails to figure out the required latent factors on given tasks. Moreover, although GPIM has a similar performance with the other three methods on 2D navigation task, GPIM has better interpretability to behaviors. As shown at the bottom of Fig. 12, considering a series of goals from the first red to the last green (*left*) in a counterclockwise order, GPIM can successfully disentangle them and learn effective latent $z$ (*right*), but *w/o KL* fails to keep the original order of goals (*middle*).

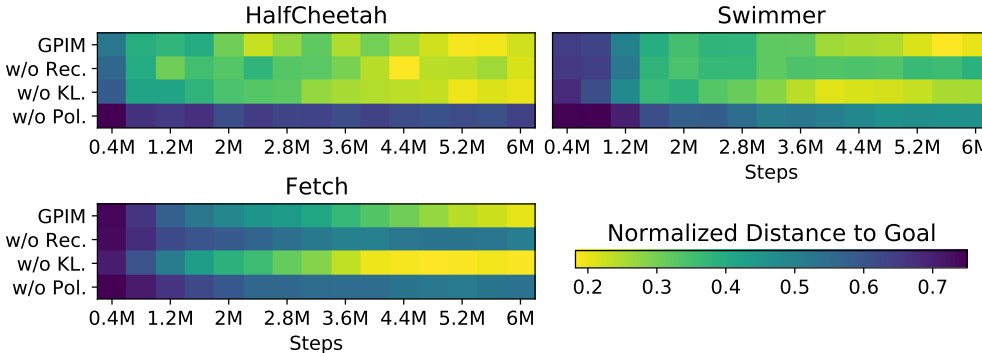

Figure 13: Ablation study on mujoco tasks.

**Ablation study on mujoco tasks.** We further study the impact of disentanglement part in our framework. The performance is reported in Fig. 13. It is observed that *w/o Pol* performs worse than all other methods, which is consistent with the performance in 2D navigation task[6]. And we can find that when we remove the reconstruction loss ($\alpha = 0$), the performance of *w/o Rec* degrades in these three environments. The main reason is that the process of learning generative factors become more difficult without the supervised reconstruction loss. While in 2D navigation task, the reconstruction loss has little impact on the performance. Even though that *w/o KL* has a similar performance with our full GPIM method, GPIM demonstrates better interpretability to behaviors as shown in Fig. 10(c).

---

[6]The results of 2D navigation task are shown in the full paper.

### A.4 GENERALIZATION ON THE GRIDWORLD TASK

Here we introduce an illustrative example of gridworld task and then show the generalization when the dynamics and goal conditions are missing in the gridworld task.

**Illustrative example.** We start with a simple gridworld example: the environment is shown on the left of Fig. 14, where the goal for the agent is to navigate from the middle to the given colored room. By abstract-level training, our method quickly acquire four skills to reach different rooms. Each time the agent arrives in a room induced by $\pi_\mu$, we train $\pi_\theta$ conditioned on the room's color (e.g., green), allowing the agent to be guided to the same room by the current reward function.

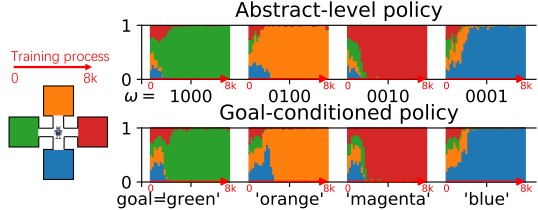

Figure 14: Gridworld tasks. Y-axis is the percentage of different rooms that the robot arrives in.

The learning process is shown on the right of Fig. 14. It is concluded that the agent can automatically learn how to complete tasks given semantic goals in an unsupervised manner.

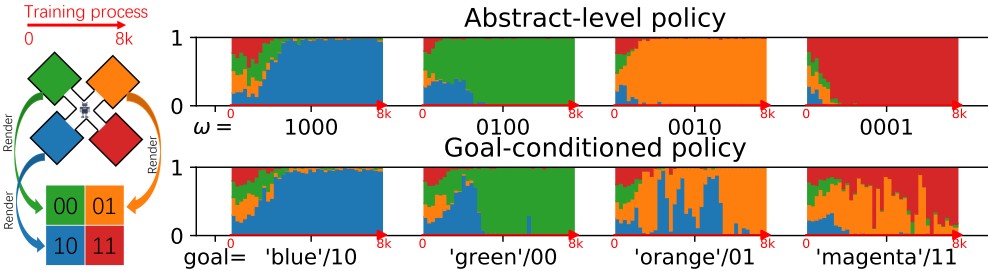

Figure 15: Gridworld tasks. The Y-axis is the percentage of different rooms that agent arrives in.

**Generalization on the gridworld task.** We further show the generalization when the dynamics and goal conditions are missing in the gridworld task, where there are four rooms in different colors: blue, green, orange and magenta. We consider the situation where we train the goal-conditioned policy in the blue room, the green room and the orange room, and test in the magenta room.

As shown in Fig. 15, we quickly acquire four skills for different rooms through the training at the abstract level. After the agent reaches at the colored rooms, we rendered the corresponding room as a two-bit representation: $Render('blue') = 10$, $Render('green') = 00$, and $Render('orange') = 01$ ; we do not render the magenta room. We take the magenta room corresponding to $11$ as the test goal to verify the generalization ability.

In lower part of Fig. 15, we show the learning process of the goal-conditioned policy. We can find that the blue room tasks are learned quickly, and the green and magenta room tasks are learned relatively slowly, but the agent is still able to complete the test task successfully (the magenta room). Compared to the task in Fig. 14 that renders all the rooms as goals, the whole learning process in Fig. 15 is much slower. We hypothesize that the main reason is that the agent needs to further infer the relationship between different goals. The lack of goal information (i.e., missing magenta $11$) leads to a lower efficiency and unstable training.

## B IMPLEMENTATION DETAILS

### B.1 DERIVATION OF THE VARIATION BOUND

$$
\begin{aligned}
& \mathbb{E}_{\omega,s,\tilde{s}} \left[\log p(\omega|s) + \log p(\omega|\tilde{s}) - 2\log p(\omega)\right] \\
= \ & \mathbb{E}_{\omega,s,\tilde{s}} \left[\log q_\phi(\omega|s) + \log q_\phi(\omega|\tilde{s}) - 2\log p(\omega)\right. \\
& \qquad \left. + KL(p(\omega|s)||q_\phi(\omega|s)) + KL(p(\omega|\tilde{s})||q_\phi(\omega|\tilde{s}))\right] \\
\geq \ & \mathbb{E}_{\omega,s,\tilde{s}} \left[\log q_\phi(\omega|s) + \log q_\phi(\omega|\tilde{s}) - 2\log p(\omega)\right]
\end{aligned}
$$

## B.2 Environment Details

We introduce the details of environments and tasks here, including the environment setting of 2D navigation (*x-y goal* and *color-shape goal*), object manipulation, three atari games (seaquest, berzerk and montezuma revenge), and the mujoco tasks (swimmer, half cheetah and fetch).

**2D navigation tasks:** In 2D navigation tasks, the agent moves in each of the four cardinal directions, where the states denote the 2D location of the agent. We consider the following two tasks: moving the agent to a specific coordinate named *x-y goal* and moving the agent to a specific object with certain color and shape named *color-shape goal*.

- **2D Navigation (*x-y goal*):** The size of the environment is $10 \times 10$ (continuous state space) or $7 \times 7$ (discrete state space). The state is the location of the agent, and the goal is the location of the final location.
- **2D Navigation (*color-shape goal*):** The size of the environment is $10 \times 10$. The state consists of the locations of the agent and three objects with different color-shape pairs (one real target and two distractors). The goal is described by the color and shape of the real target, encoded with one-hot.

**Object manipulation:** More complex manipulation considers a moving agent in 2D environment with one block for manipulation, and the other as a distractor. The agent first needs to reach the block and then move the block to the given location, where the block is described using color and shape. The size of the environment is $10 \times 10$. The state consists of the locations of the agent and two blocks with different color-shape pairs (one real target and one distractor). The goal consists of the one-hot encoding of the color-shape of the target block that needs to be moved, and the 2D coordinate of the final location of the movement.

Table 1: The repetition length of the action.

| Environments | k |
|:---:|:---:|
| Seaquest-ram-v0 | 2, 3, 4, 5 |
| Berzerk-ram-v0 | 34, 36, 38, 40 |
| MontezumaRevenge-ram-v0 | 2, 3, 4, 5 |

**Atari games:** We test the performance on three atari games: seaquest, berzerk, and montezuma revenge. In order to reduce the difficulty of training, we adopt the RAM-environment (i.e., Seaquest-ram-v0, Berzerk-ram-v0, and MontezumaRevenge-ram-v0), where each state represents a 128-dimensional vector. Each action repeatedly performs for a duration of k frames, where k is uniformly sampled from Table 1.

**Mujoco tasks:** We consider to make diverse agents to fast imitate a given goal trajectory, including the imitation of behaviors of a swimmer, a half cheetah, and a fetch, where the states in the trajectory denote the positions of agents. Such experiments are conducted to demonstrate the effectiveness of our proposed method in learning behaviors over a continuous high-dimensional action space, which is more complicated in physics than the 2D navigation.

Note that the goals in all the experiments are images $50 \times 50 \times 3$ (3 channels, RGB) in size, except that the *color-shape goal* is encoded with one-hot.

## B.3 Metrics, Network Architectures and Hyperparameters

Here we give a clear definition of our evaluation metric – "normalized distance to goal":

**I:** When the goal is to reach the final state of the trajectory induced by $\pi_\mu$ (Figs. 5(a), 5(b) and 5(c)), the distance to goal is the L2-distance between the final state $\tilde{s}_{T-1}^k$ induced by $\pi_\theta(\cdot|\cdot, g^k)$ and the goal state $g^k$ randomly sampled from the goal (task) space:

$$Dis = \frac{1}{N} \sum_{k=1}^{N} L2(\tilde{s}_{T-1}^k, g^k),$$

where $N$ is the number of testing samples. We set $N = 50$ for 2D navigation, object manipulation and atari games (seaquest, berzerk and montezuma revenge).

**II:** When the goal is to imitate the whole trajectory induced by $\pi_\mu$ (Figs. 5(d), 5(e) and 5(f)), the distance is the expectation of distance over the whole trajectory $\{\tilde{s}_0^k, \tilde{s}_1^k, ..., \tilde{s}_{T-1}^k\}$ induced by $\pi_\theta(\cdot|\cdot, \{g_0^k, g_1^k, ..., g_{T-1}^k\})$ and goal trajectory $\{g_0^k, g_1^k, ..., g_{T-1}^k\}$ randomly sampled from the trajectory (task) space:

$$Dis = \frac{1}{N}\sum_{k=1}^{N}\left(\frac{1}{T}\sum_{t=0}^{T-1}L2(\tilde{s}_t^k, g_t^k)\right),$$

where $N$ is the number of testing samples. We set $N = 50$ for mujoco tasks (swimmer, half cheetah and fetch) .

The term "normalized" means that the distance is divided by a scale factor.

Note that, for three atari games (seaquest, berzerk, and montezuma revenge), the L2-distance for evaluation[7] is the difference between the position of controllable agent and the target's position, where the position is obtained by matching the pixel on the imaged state. See the Python code below.

```python
def obtainxy_seaquest(img):
  temp = np.where((img[:,:,0]==187)&(img[:,:,1]==187)&(img[:,:,2]==53))
  global seaquestxy
  if len(temp[0])==0:
    temp = seaquestxy
  else:
    seaquestxy = temp
  xy = np.array([np.mean(temp[0]), np.mean(temp[1])])/100
return xy

def obtainxy_montezuma_revenge(img):
  img[:20] = 0
  temp = np.where((img[:,:,0]==200)&(imgo[:,:,1]==72)&(img[:,:,2]==72))
  global montezumaxy
  if len(temp[0])==0:
    temp = montezumaxy
  else:
    montezumaxy = temp
  xy = np.array([np.mean(temp[0]), np.mean(temp[1])])/100
return xy

def obtainxy_berzerk(img):
  temp = np.array(img)
  temp[1:-1, 1:-1] = temp[:-2, :-2]/3+temp[2:,2:]/3+temp[1:-1,1:-1]/3
  temp = np.where((temp[:,:,0]==240)&(temp[:,:,1]==170)&(temp[:,:,2]==103
                                                          ))
  global berzerkxy
  if len(temp[0])==0:
    temp = berzerkxy
  else:
    berzerkxy = temp
    xy = np.array([np.mean(temp[0]), np.mean(temp[1])])/100
  return xy
```

In our implementation, we use two independent SAC architectures (Haarnoja et al., 2018) for abstract-level policy $\pi_\mu$ and goal-conditioned policy $\pi_\theta$. We find empirically that having two networks share a portion of the network structure will degrade the experimental performance. We adopt universal value function approximates (UVFAs) (Schaul et al., 2015) for extra input (goals). For the abstract-level policy $\pi_\mu$, to pass latent variable $\omega$ to the Q function, value function, and policy, as in DIAYN, we simply concatenate $\omega$ with the current state $s_t$ (and action $a_t$). For goal-conditioned

---

[7]The reward function for our baseline **L2 Distance** still calculates the L2-distance directly on the original state space, instead of the distance of the agents' positions after pixel matching here.

policy $\pi_\theta$, we also concatenate $g_t$ with current state $\tilde{s}_t$ (and action $a_t$). We update the $\vartheta_E$ using the gradients from both the *Dis_loss* and Q function's loss of the goal-conditioned policy $\pi_\theta$.

The hyper-parameters are presented in Table 2.

## C  BROADER IMPACT

The main concern in the research area of goal-conditioned policy is how to find numerous diverse goals as well as obtain the reward function. Particularly, in practical application, the goal and state are often heterogeneous data with high variability of data types and formats, which further aggravates the difficulty of policy learning. Our model addresses these problems by allowing agents to interact with objects or environment, and learn the behaviors in a fully autonomous way on the basis of intrinsic motivations. Bridging with the render function between abstract level and goal-conditioned policy, we could obtain diverse and versatile policies. By optimizing the intrinsic reward, our GPIM makes an agent capable of solving diverse tasks, which fits for wide-range applications such as quick imitation on behavior states, interactive navigation and object manipulation, and so on.

Moreover, our model provides an approach to address the Robot open-Ended Autonomous Learning (REAL). This framework is likely to speed up the progress of the general-purpose robots that can achieve complex tasks given the corresponding goals, and drive the development of autonomous robot learning in a life-long learning mode.

By autonomous exploration of the environment, the agent is likely to generate some useful behaviors as well as the majority of useless skills. Of particular concern is that the use of autonomous exploration is likely to generate a strategy that will induce dire consequences, such as a collision skill in an autonomous driving environment. How to generate useful behaviors for user-specified tasks and how to use these induced skills are also open problems. An alternative solution is to use an extrinsic reward to guide the exploration or the offline reinforcement learning (Levine et al., 2020). While another issue comes from the exploration-exploitation trade-off. We would encourage further work to understand the limitations of REAL interacting with the environment autonomously. We would also encourage research to understand the risks arising from autonomous robot learning.

Another limitation is that our learning framework needs an extra abstract-level policy training for rendering goals and providing a reward function. This requires twice as much interaction time with the environment as learning a single policy network. For practical application, a simulation platform is preferred for pre-training. We also encourage researchers to mitigate the difference between the simulation platform and the actual environment. We also pursue the effective models of transfer learning.

## D  MORE RESULTS

### D.1  LEARNED BEHAVIORS ON TEMPORALLY-EXTENDED TASKS

More experimental results are given in Fig. 16 to show the imitation on several temporally-extended tasks.

### D.2  LEARNED BEHAVIORS FROM GPIM

More experimental results are given in Fig. 17 to show the learned behaviors on 2D navigation, object manipulation, three atari games (seaquest, berzerk and montezuma revenge), and the mujoco tasks (swimmer, half cheetah and fetch). Videos are available under https://sites.google.com/view/gpim.

Table 2: Hyper-parameters

| Hyper-parameter | | | value |
|---|---|---|---|
| Batch Size | | | 256 |
| Discount Factor | | | 0.99 |
| Buffer Size | | | 10000 |
| Smooth coefficient | | | 0.05 |
| Temperature | | | 0.2 |
| Learning Rate | 2D Navigation | x-y goal | 0.001 |
| | | color-shape goal | 0.001 |
| | Object Manipulation | | 0.001 |
| | Mujoco tasks | Swimmer | 0.0001 |
| | | HalfCheetah | 0.0001 |
| | | Fetch | 0.0001 |
| | Atari games | Seaquest | 0.0003 |
| | | Berzerk | 0.0003 |
| | | Montezuma Revenge | 0.0003 |
| Path Length | 2D Navigation | x-y goal | 20 |
| | | color-shape goal | 20 |
| | Object Manipulation | | 20 |
| | Mujoco tasks | Swimmer | 50 |
| | | HalfCheetah | 50 |
| | | Fetch | 100 |
| | Atari games | Seaquest | 25 |
| | | Berzerk | 25 |
| | | Montezuma Revenge | 25 |
| Hidden Size | 2D Navigation | x-y goal | 128 |
| | | color-shape goal | 128 |
| | Object Manipulation | | 128 |
| | Mujoco tasks | Swimmer | 256 |
| | | HalfCheetah | 256 |
| | | Fetch | 256 |
| | Atari games | Seaquest | 256 |
| | | Berzerk | 256 |
| | | Montezuma Revenge | 256 |
| Dimension of Generative Factor | 2D Navigation | x-y goal | 2 |
| | | color-shape goal | 2 |
| | Object Manipulation | | 4 |
| | Mujoco tasks | Swimmer | 3 |
| | | HalfCheetah | 4 |
| | | Fetch | 4 |
| | Atari games | Seaquest | 16 |
| | | Berzerk | 16 |
| | | Montezuma Revenge | 16 |
| $\alpha$ | | | 1 |
| $\gamma$ | | | 5 |
| $\delta_{pixel}$ | | | 255 |

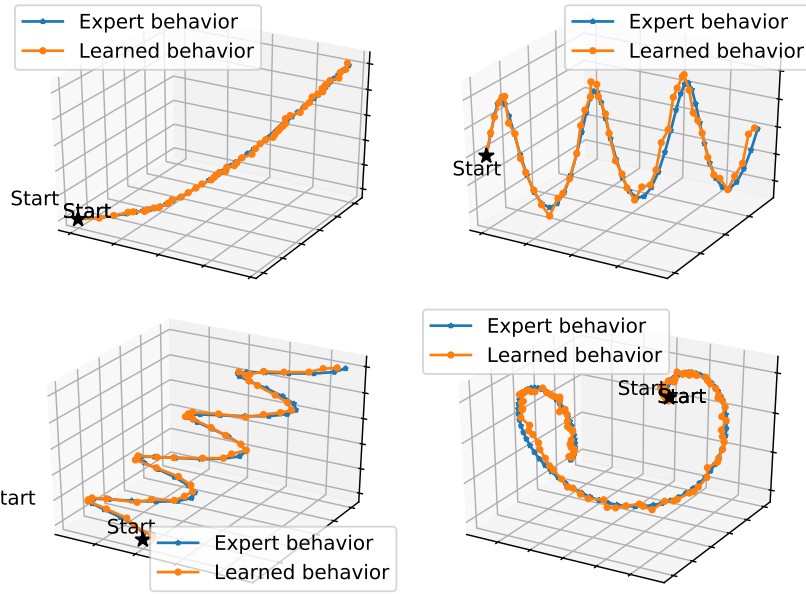

Figure 16: Expert behaviors and learned behaviors. The four expert trajectories are described parametrically as: $(x, y, z) = (\log_{10}(t + 1) + t/50, \sin(t)/5 + t/5, t/5)$, $(x, y, z) = (t/5, \cos(t)/5 - 1/5 + t/5, \sin(t)/5)$, $(x, y, z) = (\cos(t)/5 + t/50 - 1/1, \sin(t)/5 + t/5, t/5)$, and $(x, y, z) = (\sin(t) - sin(2t)/2, -t/5, \cos(t)/2 - cos(2t)/2)$.

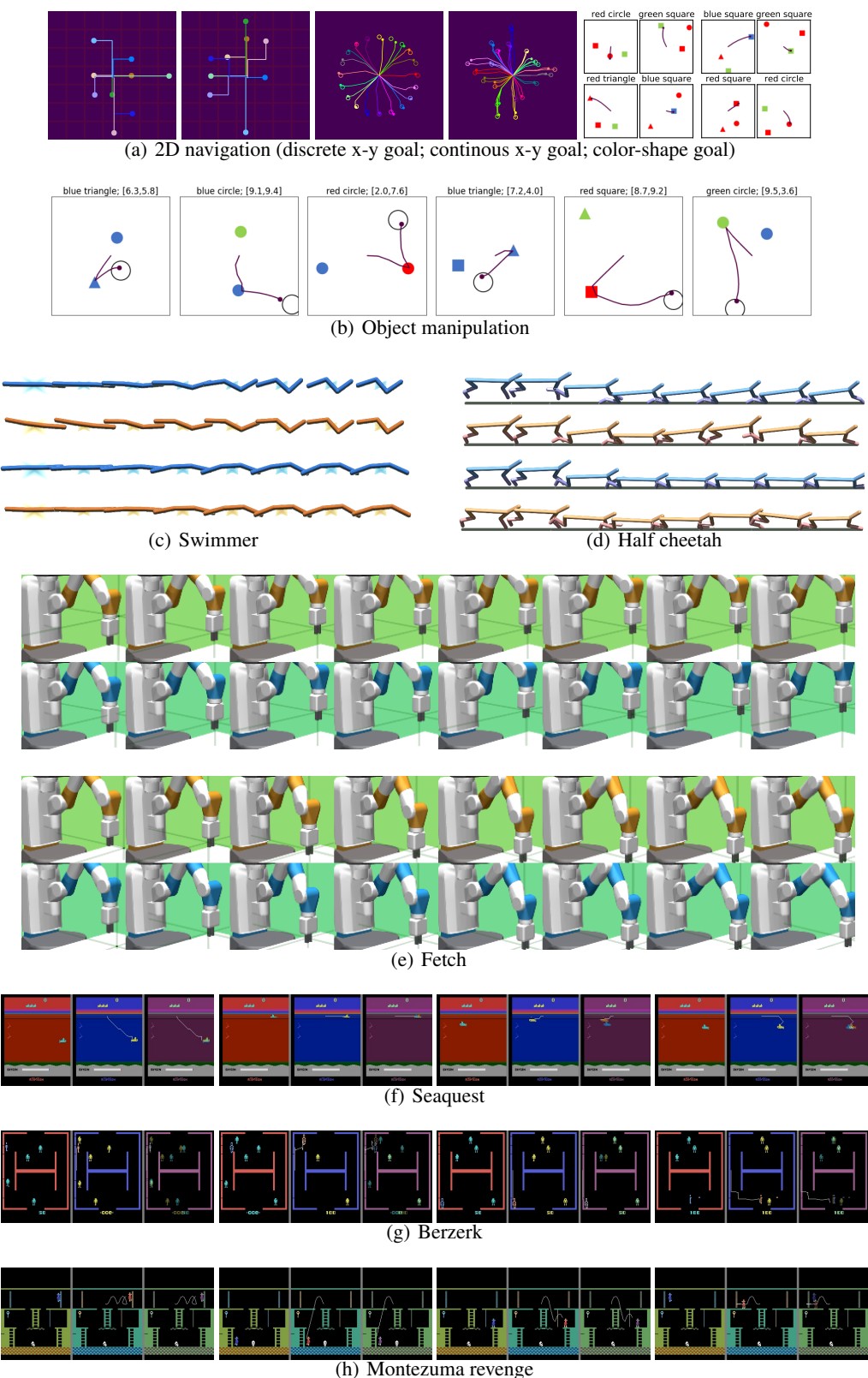

(a) 2D navigation (discrete x-y goal; continous x-y goal; color-shape goal)

(b) Object manipulation

(c) Swimmer

(d) Half cheetah

(e) Fetch

(f) Seaquest

(g) Berzerk

(h) Montezuma revenge

Figure 17: Discovered goal-conditioned behaviors. (f-h): The left subfigure shows the expert behaviors (goals); The middle subfigure shows the learned behaviors by GPIM; The right subfigure is the stacked view of goals and GPIM behaviors.

