# OpenReview forum: "Learn Goal-Conditioned Policy with Intrinsic Motivation for Deep Reinforcement Learning"
_ICLR.cc/2021/Conference — Reject_

### Official Review · AnonReviewer2 · 2020-10-27
**Novel combination of intrinsic motivation and goal-conditioned RL, assuming access to a "goal renderer"**

**Rating:** 6
**Confidence:** 4

**Review:**

This paper proposes a method for combining intrinsic motivation on a state space with goal-conditioned reinforcement learning (GCRL), where goals are defined in some “perceptual space,” such as text or images, which describe the current state. The authors assume access to a renderer that maps states to perceptual goals, but do not assume that the renderer is differentiable. The authors propose to train an intrinsically motivated latent-conditioned policy, using similar techniques as past work in which a policy maximizes the mutual information between a latent variable and the current state. The goal-conditioned policy is then trained to effectively imitate the latent-conditioned policy by maximizing the same reward as the latent-conditioned policy, conditioned on only the rendered version of the final state reached by the latent-conditioned policy. The authors demonstrate that the overall method outperforms past GCRL methods on a variety of tasks (Atari, MuJoCo manipulation and locomotion, and toy tasks).

The idea of using intrinsic motivation rewards to train a GCRL policy is an interesting and novel idea, and the experiments indicate that this is a promising approach. The writing is a bit verbose and there are some details that rushed through, but overall I find the idea novel and the results compelling.

I would recommend that the authors clarify that this method assumes access to a renderer. Currently, the writing does not make it clear that the comparison to RIG and DISCERN is not an apples-to-apples comparison: RIG and DISCERN do not assume access to the ground-truth state, but rather operate directly in the perceptual space. However, I think this comparison is reasonable since I do not know of any other method that makes the same assumption as this paper.

The authors state that, “the goals for πθ in mujoco tasks are the rendered trajectories induced by π...” In that case, how did the authors condition a policy on entire trajectories? This seems like an important detail, and it’s also unclear how the baselines could have been implemented in those cases.

The analysis of Figure 6 experiment seems to overstate the result. While it is true that, “during training the agent is required to imitate a large number of simple behaviors” it does not seem to be the case that it “has never seen such complex goals.” Unless I am mistaken, the policy at test time is not conditioned on a single complex goal at test time. Instead it’s simply conditioned sequentially on simple tasks, which is effectively the same as during training but just done sequentially.

--- Post Rebuttal ---

I've read the author response and do not intend to increase my score. Thank you for answering my questions. It would be good to clarify in the main paper that conditioning on the trajectory is implemented in the way described in the author response.

---

> ### Author Response · Authors · 2020-11-16
> **Clarification: render; policy conditioned on a trajectory;  sequencial simple tasks**
>
> Thank you for commenting in great details and finding our motivations and studies valuable.
>
> Q1: access to a render
>
> A1: In RL, especially in the area of domain randomization, plenty of methods assume access to the render, leading to a redundant set of easy data for learning. For the goal-reaching tasks, from the 2D navigation to atari games, and from mujoco tasks to real robots, it is easy to obtain the ground-truth state and the visual/sensor input. Therefore,  it is straightforward and important to explore the different characteristics of them.
>
>
> Q2: How did the authors condition a policy on entire trajectories? How the baselines could have been implemented in this case.
>
> A2: In the case that the goals for $\pi_\theta$ in mujoco tasks are the rendered trajectories induced by $\pi_\mu$, the goals for $\pi_\theta$ are dynamic, e.g., $\pi_\theta(a_1 | s_1, g_1)$ --> $\pi_\theta(a_2 | s_2, g_2)$ --> $\pi_\theta(a_3 | s_3, g_3)$ ... , not $\pi_\theta(a_t | s_t, g_1, g_2, g_3, ...)$. For these dynamic goals, the baselines only need to provide time-varying reward signals: $r_t = r(s_t, a_t, g_t) = r(s_{t+1}, g_t)$. The baselines adopt the prior non-parametric distance function to obtain the reward over the learnt embedding space, so it is straightforward to implement.
>
>
> Q3: the experiment in Figure 6: a single complex goal vs. simple tasks that are sequenced.
>
> A3: A single complex goal and sequential simple tasks are themselves ambiguous. One critical element of solving temporally-extended tasks is the notion of compositionality. It is true that we test our model on sequential simple tasks. Our model shows the compositional ability, enabling fast inference and combinatorial generalization.

---

### Official Review · AnonReviewer1 · 2020-10-28

**Rating:** 7
**Confidence:** 4

**Review:**

Summary
--
This paper proposes an unsupervised learning objective for learning perceptual goal-conditioned policies. The goal is to enable unsupervised discovery of high-level behaviors in tandem with a perceptual-goal conditioned policy that can achieve these behaviors. The learning proceeds by training one policy to exhibit diverse behaviors; the states induced by these behaviors are then rendered and used as target goal states for a separate goal-conditioned policy.

The learning objective is to maximize a lower-bound on sum of two terms -- (1) the mutual information between the behavior variable and states with a behavior-conditioned policy; states from this behavior-conditioned policy are transformed into perceptual goals that serve as input to a separate policy that forms (2) the mutual information between perceptual goals and the states it induces.

The learning algorithm operates via alternating optimization, in which the skill-conditioned exploration policy is learned jointly with a skill 'discriminator' (inference network), and then the perceptual-goal conditioned policy is learned using the discriminator as a reward signal, which essentially estimates the extent to which a robot is achieving a skill given a perceptual goal along the skill-policy trajectory.

A glut of experiments are used to investigate whether the method learns a meaningful skill reward function, whether it can achieve goals in various environments, how the method compares to related methods, and whether the specific 'disentanglement' inductive bias for constructing the policy is useful. The experiments demonstrate favorable performance over existing methods for these tasks.

Quality
--
The goal, method, and experiments are likely high quality, given my understanding. However, there are significant gaps in clarity that reduce my certainty in this assessment, and relatedly, there is some important missing discussion on the specific differences between the proposed method and prior work.

Clarity
--
- The learning algorithm is ambiguous: which goal(s), specifically, are used in the second learning stage? The current algorithm reused the 't' time index, which makes this unclear. I suspect it's just the goal corresponding to the last timestep from the first stage of the algorithm, but I'm not sure.
- The state representation for the archery task is unclear
- The 'fast imitation' procedure is unclear. The paper says the goals are the rendered states induced by the abstract policy, so how do the expert demonstrations get factored in? Is the learning algorithm different (is the abstract policy trained with the expert demonstrations somehow)? This is a significant ambiguity that makes it difficult to interpret the results of the imitation experiments. It's hard to guess at, because the conceptual difference between the standard imitation learning problem and a prototypical unsupervised "RL" algorithm is quite large.
- The presence of two environments in Figure 1 is quite confusing. It's hard to tell from the context of the figure alone whether the policies are operating simultaneously in the same environment, simultaneously in separate equivalent environments, or at separate times in equivalent environments.
- The \tilde notation and relationship between the \tilde an not-\tilde variables needs to be discussed in 3.1

Originality
--
The learning objective seems to be quite similar to the DIAYN and DISCERN objectives, and the task of learning to condition on general perceptual goal in unsupervised setting is shared by RIG and DISCERN. Discussion of specific algorithmic and assumption differences between the proposed method and these approaches is quite necessary, but unfortunately missing.

Significance
--
In absence of knowing more about the specific differences between the proposed work and related works and the imitation learning experiments, the only thing that is clear is that the method seems relatively performant on an existing well-motivated task across a large range of settings, and compares favorably to existing methods on this task.

Other points
--
Derivation is needed (e.g. in appendix) to show that (3) is a lower bound of the last line of (2). I don't think this aspect of the derivation follows from Jensen's inequality, as stated in the paper.
- In the intro (para 3) "an unsupervised method in RL" is unclear to an RL reader unfamiliar with "unsupervised RL". I think this paper should explicitly define "unsupervised RL" to mean "RL with an intrinsic reward function" or something equivalent, and "intrinsic reward function" to mean a learning objective that can be applied in place of an alternative reward function across different MDPs.
- Intro para 3 "maximize an information theoretic objective" is vague, because all learning objectives for probabilistic models are technically "information theoretic" -- all probabilistic models have a relationship to information.

---

> ### Author Response · Authors · 2020-11-16
> **Thanks for your time!   :)**
>
> We thank the reviewer for the thoughtful comments. Our replies to the major concerns are as below:
>
>
> Q1: Which goal(s) are used in the second learning stage?
>
> A1: In our experiment, we consider the two cases: (1) Case I: render induced state at the last time step as a fixed goal for the goal-conditioned policy, and (2) Case II: render the whole trajectory as dynamic goals for the goal-conditioned policy (imitating expert trajectories).  See the last few lines of the first paragraph of the subsection "Scaling to more complex tasks".
>
>
> Q2: the state of archery.
>
> A2: the position of the arrow.
>
>
> Q3: The "fast imitation" procedure is unclear.
>
> A3: The "fast imitation" refers to the validation stage, not the training phase. In fact, we hope the agent can achieve zero-shot imitation of new expert trajectories after it learns to imitate diverse trajectories induced by the abstract-level policy.
>
>
> Q4: The presence of two environments in Figure 1 is confusing.
>
> A4: We thank the reviewer for this comment. We should have clarified this. These two environments are the same.
>
>
> Q5: The \tilde notation and relationship between the \tilde and non-\tilde variables need to be discussed in 3.1.
>
> A5: We thank the reviewer for the suggestion. We have added a description of the notation and the relationship on the newly updated paper.
>
>
> Q6: discussion difference between our method, DIAYN, DISCERN, and RIG.
>
> A6: We have added additional experiments to compare with DIAYN and its variants in the Appendix. The differences between our model and baselines (DISCERN and RIG) contain two parts: (1) a mechanism for automatically generating goals for goal-conditioned policy, and (2) the expressiveness of the learned reward functions. We have shown that our straightforward framework can effectively explore the environment, achieving competitive performance with baselines (see appendix A.2 (the updated version).
>
> In this rebuttal phase, we have added new experiments to compare the expressiveness of the reward functions between different methods. We refer the reviewer to the updated paper.
>
>
> Q7: the derivation of Eq. (3).
>
> A7: We have added the derivation in Appendix B.1.
>
>
> Q8: description -- "An unsupervised method in RL" and "maximizing an information theoretic objective"
>
> A8: We have added more descriptions.

---

> > ### Comment · AnonReviewer1 · 2020-11-20
> > **Response**
> >
> > I thank the authors for addressing my concerns and improving the clarity. The addition of the experiments displayed in Fig. 9 is helpful. As mentioned by me and other reviewers, comparison to DIAYN, DISCERN, and RIG was needed. I admit I'm a bit confused by the response stating these new results are in A.2 (there's no new "red" text in A.2); Fig. 9 and its surrounding discussion seems to contain these new results..?
> >
> > As a result, I'll increase the confidence in my existing positive assessment. However, I do share R1's opinion that "some parts of the writing ... seem out-of-place at first". I think that there is quite a lot going on in the paper, and that the writing could be globally streamlined and the main points distilled.

---

> > > ### Author Response · Authors · 2020-11-21
> > > **New "red" text in A.2**
> > >
> > > Dear Reviewer,
> > > Thank you for the feedback and review. We have been constantly revising the structure and content of our manuscript. We have revised our manuscript in the introduction according to your comments.
> > >
> > > Q1: New "red" text in A.2
> > >
> > > A1: Sorry for the confusion. The "the updated version" we mentioned refers to the structure of the appendix, not the content. The content is the same as before.
> > >
> > > P.S.: We kindly remind the reviewer that your reviewer number is #1, and the first reviewer's number on the page is #4.

---

### Official Review · AnonReviewer3 · 2020-10-30
**Like the Extensive Experiments, But The Paper Requires Plenty of Clarification**

**Rating:** 6
**Confidence:** 3

**Review:**

Summary: The paper proposes a novel method for learning goal-conditioned policy with images/text goals.

Quality: The overall quality of the paper is good.
    Strong side: Extensive evaluation on various environments and tasks showcases the advantage and generalizability of the method. The authors uses figures and algorithm boxes to make their method very clear.
    Weak side: Several clarification on the motivation and details of the method is needed. See below.

Clarity: (1) The main argument is not strong: 1. Image/text goals is not intractable for current goal-conditioned policies [1] 2. What do you mean by ‘extrinsic’ reward? If you mean task-specific reward, many methods learn goal-conditioned policy without extrinsic reward, like hindsight experience replay.
(2) Therefore, I guess the authors wanted to claim they use ‘intrinsic’ rewards, which is a mutual-information-based reward. Now I have two questions for Section 3.2: 1. Why do we use this loss function Eq. (1)? It comes out of nowhere without intuition. (2) Why do you decompose the optimization into ($\mu, \Phi$) and $\theta$? I feel like decomposing into $\Phi$ and ($\mu, \theta$) is more reasonable in that one optimizes the reward first then policies.
Another question on algorithm box: In step two, $g_t$ is changing with time steps. This is kind of strange because in typical goal-conditioned policy learning, in one epoch, people use a fixed goal to train their policy, then change the goal in the next epoch. What is the fundamental difference?
(3) The whole disentanglement thing needs more clarification. I don’t understand the reason why you disentangle your policy. In computer vision, disentanglement has clear physical meanings like disentangling shape and color, but here I don’t have such intuition. In experiments, the effect of disentanglement is only demonstrated in a simple 2D-task, which seems not enough.
(4) Details: 1. How is p(w) defined? It seems super important, but I was not able to find its details in the paper. Maybe I overlooked something.

Originality: As far as I know, the method is new.

Significance: Learning goal-conditioned policy with high-dimensional goals is an important problem. I think if the authors could clarify the above questions, the solid experiments will make the paper a good contribution. However, I don’t think its current version passes the bar of ICLR.

Reference: [1] https://papers.nips.cc/paper/9623-planning-with-goal-conditioned-policies.pdf

---

> ### Author Response · Authors · 2020-11-16
> **significant revision to our manuscript**
>
> We appreciate the reviewer's informative feedback. We will address each of your major points in the sections below, following your questions/comments.
>
>
> Q1: main argument: image/text goals are not tractable for current goal-conditioned policies.
>
> A1: This is not our main argument. Our argument is that the training of abstract-level policy can provide diverse goals and valid reward signals for the goal-conditioned policy (which is conditioned on the generated goals). The underlying manifold spaces of skill ($\omega$) and goal (g) can naturally avoid the issues raised from the heterogeneous goal and state.
>
>
> Q2: "extrinsic reward" in the introduction.
>
> A2: we refer to the prior reward function, e.g., the non-parametric distance function in original or embedding space. We have modified the description.
>
>
> Q3: Why do we use the loss function in Eq.(1): $I(s; \omega) + I(\tilde{s}; g)$.
>
> A3: The first term is to maximize the mutual information between skills and states, I(s; $\omega$), to encode the idea that the skill should control which states the agent visits. This objective can result in the unsupervised emergence of diverse skills, such as moving in a different direction. This is consistent with DIAYN. The second term is similar in spirit to DISCERN, which maximizes the mutual information between the achieved final state $s_T$ and the goal state $s_g$: $I(s_T; s_g)$. We maximize the mutual information of the whole trajectory and the goals.
>
>
> Q4: Why do we decompose the optimization into ($\mu$, $\phi$) and $\theta$?
>
> A4: We speculate that the description -- "The abstract-level policy $\pi_\mu$ and goal-conditioned policy $\pi_\theta$ share an identical reward network (i.e., discriminator) $q_\phi$ " results in this concern. In fact, we first update the abstract-level policy $\pi_\mu$ and the reward function $q_\phi$, then the learned $q_\phi$ can serve the reward function for goal-conditioned policy $\pi_\theta$. We do not use the data generated by $\pi_\theta$ to update discriminator $q_\phi$. In other words, $\pi_\mu$ and $\pi_\theta$ are not parallel. The abstract-level offers reward function $q_\phi$ and generates goals for training goal-conditioned policy $\pi_\theta$. We have made significant revisions to our manuscript because of this misunderstanding.
>
>
> Q5: $g_t$ is changing with time steps vs. $g_t$ is fixed.
>
> A5: In RL, we think these two cases are consistent, as long as we can provide the reward signal at the corresponding time step. In our experiment, we consider the two cases. See the last few lines of the first paragraph of the subsection "Scaling to more complex tasks".
>
>
> Q6: the intuition of the disentanglement.
>
> A6: Note that we disentangle the high-dimensional input, not the policy itself. For high-dimensional input for the goal-conditioned policy, the disentanglement using an information bottleneck can be used to improve generalization. Our motivation is consistent with the area of computer vision. For example, the 2D navigation task shows the disentanglement of the color and shape of the object.
>
>
> Q7: How is $p(\omega)$ defined?
>
> A7: We use a categorical distribution for $p(\omega)$ following DIAYN.
>
>
> We have made significant revisions to our manuscript, and we hope that our updates to the manuscript address the reviewer's concerns about clarity.

---

> > ### Comment · AnonReviewer3 · 2020-11-22
> > **Reply to the Authors**
> >
> > The current version is much clearer than the previous one. Now I feel the method is well-motivated and reasonable. The visualization of the learning process is also very helpful. I will increase my score to a 6.
> >
> > However, I still feel like the disentangle part is not very relevant to the other parts of the method. I suggest the authors to  conduct more controlled experiments to justify its usefulness.

---

### Official Review · AnonReviewer4 · 2020-10-30
**Weak evaluation**

**Rating:** 5
**Confidence:** 3

**Review:**

# Summary
This paper proposes a new solution to the problem of learning goal-conditioned policies without hand-crafted rewards. Prior work in this domain learn an embedding space to compute reward between current state and goal. In contrast, this paper utilizes unsupervised skill discovery from [1] to obtain a discriminator that identifies which states belong to a particular skill. Then, the final state of a given skill's execution is used as a goal input to a goal-conditioned policy, which is rewarded if it generates states that the discriminator identifies with this skill. The paper aims to validate the benefit of such a reward over other embedding-distance based reward functions on a variety of environments.
​
​
----------
​
# Strengths
- The paper solves an important problem of unsupervised learning of goal-conditioned policies. To the best of my knowledge, the proposed way of training a goal-conditioned policy with skill-based discriminator is novel.
- The training framework combines unsupervised exploration (skill-discovery) and learning of goal-conditioned policy elegantly through the discriminator. I find the insight very interesting that the learned skill's end-point can be treated as a target goal and the skill discriminator can be used as a reward to identify the states that are going towards the goal (since these states fall on the skill's induced trajectory).
- The experiments are conducted on a large set of environments, which is good to justify the claims made by the paper.
- I appreciate the video results and the visualizations provided on the project website.
​
​
----------
​
# Weaknesses
- The paper is missing the ablation study for the primary contributions. The paper argues that approaches taking reward as distance in embedding space can limit the repertoires of behaviors. One such example of unsupervised reward function is DISCERN [2]. To validate this point, and to make a fair comparison with DISCERN, the states used to train DISCERN's reward predictor should be generated from the same abstract-level policy (from [1]) as used by this paper. The current DISCERN baseline seems to be training the reward function on random exploration in the environment. I think a stronger version of this baseline is DIAYN+DISCERN.
​
- The section about disentanglement using information bottleneck seems orthogonal to the main contributions of the paper - about an unsupervised method to train goal-conditioned policies. While the information bottleneck can always be used to improve generalization, it is unclear why its use is emphasized. If I understand this correctly, then the ablation study in Section 5 can be moved to Appendix as it seems an extra detail. It is rather more important to ablate the role of the intrinsic reward function proposed in this paper (as discussed in the above point).
​
- The writing is understandable, but can be improved. It is not always clear what the main contributions of this paper are.
(a) The "abstract-level policy" is precisely the unsupervised skill-discovery framework from [1], and it should be stated as such.
(b) The paper emphasizes the view that the exploration and goal-conditioned policies share the same reward function, but a more relevant viewpoint is that the proposed training framework enables the discriminator learned for exploration to also be useful as a reward for the goal-conditioned policy.
(c) Some parts of writing are incoherent and seem out-of-place at first. For instance, paragraph 3 of introduction should be written in a way that contrasts this paper's contributions to the prior work.
(d) One clarification question I have is about where the goals for evaluation (such as in videos) come from? Are these goals just what were observed in training or are these manually-crafted goal-locations for evalution? It would be nice if a principled evaluation procedure is used, where a dataset of testing goals (maybe procedurally generated) are used to report the performance of the goal-conditioned policy.
​
​
----------
​
# Reason for decision
My primary concern is the missing baseline where alternate unsupervised rewards are combined with better exploration methods (just like this paper uses [1]). In my understanding, the primary contribution is an alternate reward function, which does not depend on explicit embedding distances. To validate this contribution, this baselines is important. I would be happy to increase my score if this concern is addressed.
​
​
----------
​
# Suggestions for improvement
- It was a little difficult to understand the high-level intuition of the method. It can be more succinctly summarized in the text and/or Figure 1 caption, such as: "Skill w is a way to reach the goal g, and the goal-conditioned policy with goal g is rewarded to imitate the trajectory induced by w."
​
​
----------
​
# References
[1] Benjamin Eysenbach, Abhishek Gupta, Julian Ibarz, and Sergey Levine. Diversity is all you need: Learning skills without a reward function. arXiv preprint arXiv:1802.06070, 2018.
[2] David Warde-Farley, Tom Van de Wiele, Tejas Kulkarni, Catalin Ionescu, Steven Hansen, and Volodymyr Mnih. Unsupervised control through parametric discriminative rewards. arXiv preprint arXiv:1811.11359, 2018.

---

> ### Author Response · Authors · 2020-11-16
> **New experiments and significant revision to our manuscript**
>
> We thank the reviewer for the thoughtful comments. Our replies to the major critiques are as below:
>
> 1 . Missing ablation study -- "taking reward as distance in embedding space can limit the repertoires of behaviors".
>
> We thank the reviewer for the suggestion. This is indeed an insightful experiment. So, we pose a new question -- "Does the learned reward function produce better expressiveness of tasks, compared with the prior non-parametric function in the embedding space?". We answered it empirically in the subsection "Expressiveness of the reward function".  For details of this experiment and analysis, please refer to this subsection (page 8-9).
>
>
> 2 . Disentanglement using the information bottleneck is orthogonal to the main contribution.  The reviewer suggests that we move the ablation study to the appendix.
>
> We thank the reviewer for the suggestion. We have moved the ablation study to the appendix.
>
>
> 3 (a). Compared with DIAYN and DISCERN.
>
> This model is closely related to DIAYN and DISCERN. A difference is that the policy is factorized into two parts (abstract and goal conditioned) to potentially facilitate better generalization (between abstract-level and goal-conditioned level). We factorize the skill and learn it purely in the space of the agent's embodiment -- separate from the image/sensor specified goals.  The space of goals in these two spaces has different characteristics due to the underlying manifold spaces. Hence we separate them into two policies with a shared reward function.
>
>
> 3 (b). exploration and goal-conditioned policy share the same reward function.
>
> We agree with the reviewer that this might cause confusion. The exploration we emphasized takes place during the process of training abstract-level policy, and the learned discriminator serves as a reward function to train goal-conditioned policy. We have modified the description in the updated paper.
>
>
> 3 (c). Paragraph 3 should be written in a way that contrasts the paper's contribution to the prior work.
>
> We have an idea of the misunderstanding caused to the reviewer. In Paragraph 2, we analyze the issues that were raised by the prior work maximizing $I(s;g)$, and then we analyze the issues raised by maximizing $I(s;\omega)$ in Paragraph 3 for solving the goal-reaching tasks. In fact, the learned discriminator and skills in DIAYN also can be adopted to goal-conditioned tasks. We compare our model to them in Appendix A. We hope this could help the review. We will polish the introduction soon!
>
>
> 3 (d). goals for evaluation.
>
> The y-axis in Fig.7  is "Normalized distance to goals", and the y-axis in Fig.8 is "Distance to rendered goals".  The goals for evaluation in Fig.7 are not the rendered goals at the abstract-level. They are randomly sampled in the goal space (or by rolling out a new abstract-level policy which is extra trained). The goals for evaluation in Fig.10 are the rendered goals at the abstract-level.
>
>
> In all,
> Thank you for your time. We hope you find that our revision addresses your concerns.

---

### Author Response · Authors · 2020-11-22
**Manuscript updated**

Manuscript updated

Dear reviewers,
We would like to let you know that we have updated the manuscript with the changes requested in your reviews. Thank you again for your feedback.

**New results to show the expressiveness of the learned reward function**: As suggested by R4, we ran additional experiments comparing the performance of stronger baselines: DIAYN+RIG, DIAYN+DISCERN.

**Writing**: We further describe and emphasize the motivations in the introduction section and the method section.

---

### Decision · Program_Chairs · 2021-01-07
**Final Decision**

**Decision:**

Reject

**Comment:**

This work extends previous work on unsupervised learning of goal-conditioned policies: an abstract skill policy, which drives exploration of the state space, is used to propose goals as well as derive rewards for a goal conditioned policy.

Reviewers agreed the approach was novel and interesting. All reviewers raised significant concerns about clarity and/or lack of details, as well as a lack of comparison to DIAYN/DISCERN, though these points were adequately addressed in revisions. One remaining issue raised by two reviewers are that the content related to the information bottleneck/disentangled representation learning seems out of place and ill-justified. Detailed discussion of this aspect of the work has been relegated to the appendix.

This is an important problem and a growing area of study, and while the submission has potential, improvements needed are not minor, and given the short process, we can only accept papers as is, rather than expecting certain changes. We urge the authors to further improve the focus of the work and perhaps plan to investigate the role and importance of disentanglement with IB in this setting in follow up work wherein they have the space to properly do justice to the topic in its own right.